# Development of the Endo-Reproductive System and the Effect of Mating Status on Egg Development in Adult *Hermetia illucens* L.

**DOI:** 10.3390/insects16040330

**Published:** 2025-03-21

**Authors:** Xiangying Chen, Lingqiao Li, Fan Hu, Yan Wang, Yijun Zhang, Zihao Zhou, Qiong Zhou

**Affiliations:** College of Life Sciences, Hunan Normal University, Changsha 410081, China; potato@hunnu.edu.cn (X.C.); lilq1@shanghaitech.edu.can (L.L.); fan_hu_2019@aliyun.com (F.H.); wyan@hunnu.edu.cn (Y.W.); 202030153009@hunnu.edu.cn (Y.Z.); zihao_zhou@aliyun.com (Z.Z.)

**Keywords:** *Hermetia illucens* L., endo-reproductive system, anatomy, mating status, ovum development

## Abstract

The aim is to understand the development process and structural characteristics of the endo-reproductive system of *Hermetia illucens* L. and then lay a preliminary foundation for the research of its reproductive behavior and artificial breeding. Using a stereoscopic microscope and optical microscope, we investigated the development process of the endo-reproductive system of adult *H. illucens* and explored the effect of mating on the development of its eggs. The results show that the female *H. illucens* endo-reproductive system mainly consisted of ovary, spermatheca, spermathecal gland, lateral oviduct, female accessory gland, while the male *H. illucens* endo-reproductive system mainly consisted of testicle, seminal vesicle, vas deferens, ejaculatory duct, male accessory gland. The dissection results suggest that mating can promote the development of eggs in *H. illucens*. Additionally, the maturity level or mating status can be determined based on the morphology and contents of the endo-reproductive system.

## 1. Introduction

The black soldier fly (*Hermetia illucens* L.), (Diptera: Stratiomydiae), is a widely utilized saprophytic insect [1]. It is native to America but has a broad survival range (primarily between 45° N and 40° S) [2]. Currently, *H. illucens* is widely distributed in various regions in China including Beijing, Tianjin, Henan, Hebei, Shandong, Fujian, Sichuan, Yunnan, Hunan, Hubei, Guangdong, Guangxi, Hainan, Taiwan, and Hong Kong [3,4]. Many insect species with waste degradation capabilities, including *H. illucens*, have been utilized in many countries to tackle the enormous waste produced in large-scale livestock and food farming [5,6]. For example, *H. illucens* larvae have been applied to transform kitchen waste into a protein-rich source for high-quality animal feed [7]. Also, livestock manure treated with *H. illucens* not only has a high degradation rate but also effectively inhibits the number of *Escherichia coli* and *Salmonella* in the manure [8], a feature which has proved effective in treating both chicken and pig manure [9,10]. Additionally, the high oil content of *H. illucens* larvae can be utilized to extract biodiesel [11]. The high level of antimicrobial peptides and medium-chain fatty acids in larvae can be applied in biomedicine as well [12]. Further, they can also be used to extract biological active substances such as chitin, antimicrobial peptides, and fatty acids, and to refine biodiesel or other biological active substances [13,14,15]. *H. illucens* powder can be utilized as animal feed for fish and as an animal feed additive [6].

Insects, as the most diverse, abundant, and geographically widespread animal group, have strong fecundity and reproductive adaptations. Their reproductive systems are important structures for gametogenesis and continuous breeding and they are categorized into two parts: the external genitalia and the internal genitalia [16]. The development of the internal genitalia has a direct impact on their fecundity. Therefore, the structure, function, and developmental characteristics of the endo-reproductive system underlie the reproductive capacity and adaptability of insects. The developmental state of the endo-reproductive system is also directly linked to their reproductive behavior, stage of adult development, mating status, and age, which can serve as reliable indicators for insect management and rearing [17]. During insect mating, males transfer sperms to females, which causes behavioral and physiological changes in the females [18]. For instance, mating promotes ovarian development and egg-laying behavior in female individuals [19]. Secretions from male accessory glands (male accessory gland, MAG) are an important substance regulating reproduction in female insects [20], promoting egg maturation and stimulating egg laying in females [21]. Currently, most research reports on *H. illucens* concentrates on their applications, with little data available on the morphological and structural characterization of their reproductive system [22,23]. The developmental pattern, composition, and structural characteristics of the endo-reproductive system of *H. illucens* are an important foundation for studies on their mating behavior and artificial breeding. In this study, we dissected the endo-reproductive system of laboratory-reared adult *H. illucens* to describe their morphological features, as well as the development of their reproductive organs. The results provide information on the internal structure that is involved in reproductive behaviors, such as mate selection and spawning selection.

## 2. Materials and Methods

### 2.1. Sources of Test Insects

Black soldier fly pupae were provided by Ningxiang Fengyu Biotechnology Company, Ningxiang County, Hunan Province, China. The experiment was conducted at the Insect Chemistry Ecology and Behavioral Laboratory, Hunan Normal University, Changsha City, Hunan Province, China. The larvae were reared under controlled conditions of 26–30 °C and 50–70% humidity in an insect-rearing chamber. Larvae were fed on a wheat bran diet prepared at a ratio of wheat bran–water = 1:1, following Sheppard et al. [8], until they pupated. The pupae were collected and maintained at 26 °C until emergence. Emerging adults old were collected for experiments at 1–10 days.

### 2.2. Rearing Methods

Emerged *H. illucens* adults were reared following two regimes in an insect-rearing cage (30 × 30 × 30 cm) and under a temperature of 27–29 °C and moisture of 50–60%: one involved rearing the same number of males and females in the same cage together and the other involved rearing males and females separately in different cages. All insects were provided with distilled water. Eggs were collected for continuous rearing.

### 2.3. Recording of the Anatomy of the Reproductive System

Unmated male *H. illucens*, and mated and unmated female *H. illucens* from 0 to 10 days old were separately collected for morphological dissection for each sex, mating situation, and age of adults, at least 10 individuals were used. The structures of the reproductive organs were observed and photographed after dissection. Then, the developmental characteristics were recorded. Data relating to mated females were obtained from adults in the mixed cage, while data on the males and unmated females were obtained from the separate cage. The presence or absence of spermatozoa in the spermatheca of female individuals was used to determine whether or not they were mated.

Morphological dissections were performed under a stereoscopic microscope (LEICA S8APO, LEICA, Hessen, Germany). Before dissections, the *H. illucens* were frozen in a refrigerator freezer (−20 °C) for about 1 min, and then the dissection started once they had lost the ability to move. A dissecting needle was used to immobilize the *H. illucens* at the midline of their thorax, followed by the detachment of the legs and wings using Venus scissors (Suqian Shifeng Medical Equipment Co., Ltd., Suqian, China). Afterwards, the abdomen was flooded with 0.9% saline solution and an incision was made along both sides of the abdomen with the Venus scissors. The tergite and sternite were then carefully removed using forceps, while leaving the intra-abdominal tissue. The abdominal tissue was separated from the thorax by cutting at the anterior end of the alimentary canal with a Venus scissor. The intra-abdominal tissues were transferred to a saline-dripped culture dish by holding the external genitalia with forceps. The reproductive system was exposed by excluding the alimentary canal, ventral nerve cords, trachea, and other tissues, and it was subsequently rinsed multiple times with saline to eliminate the attached fat bodies. The study observed the developmental status of various organs in the reproductive system, such as the ovaries, female accessory gland, spermatheca, testicle, and male accessory gland. To capture the morphological features of the reproductive system, micrographs were taken with the LAS V4.9 software system accompanying the stereo microscope, and photos were obtained. At the same time, a distinct slide was extracted from the spermatheca in the reproductive system of the female individuals, and their fertilization was observed via an SOPTOP EX20 optical microscope (Sunny Group, Suzhou, China) (captured at a 40× magnification).

### 2.4. Data Measurement and Analysis

The dimensions of the testicle and ovary of *H. illucens* were measured by utilizing the image measurement and analysis software Digimizer Version 4.3.4. Length was taken as the distance from the base to the end of the organ and width was measured at the widest part of the organ. Statistics and analysis of variance (ANOVA) were performed for each index using the IBM SPSS Statistics 23 software. One-way ANOVA analysis and Duncan’s multiple range tests were used to test the significance of the differences in the length and width of the *H. illucens* spermatheca at different ages.

## 3. Results and Analysis

### 3.1. Endo-Reproductive System of Adult H. illucens

The endo-reproductive system of the male fly includes a pair of testicles, a pair of accessory glands, vas deferens, seminal vesicle, and ejaculatory ducts (Figure 1A). The testicles, long, thick, white or yellowish tubes, attach to the ejaculatory ducts at the base by a long, thin, white vas deferens, which partly expands into the seminal vesicle. The seminal vesicle is an irregular, thin, elongated sac, the volume of which varies to some extent at different ages. The male accessory glands are transparent, elongated, and tubular, linked at the base to the vas deferens and ejaculatory ducts. The ejaculatory ducts are slightly thicker, tubular, and are joined to the male genitalia.

The endo-reproductive system of the female fly comprises a pair of ovaries and lateral oviducts, a pair of female accessory glands, three spermathecae, and three spermathecal glands (Figure 1B). The ovaries are connected to the middle oviduct at their bases through the lateral oviducts. The size, color, and shape of ovaries vary as the age increases. Additionally, the female accessory glands are transparent, with a long tubular shape and thin bases. These accessory glands are also attached to the middle oviducts. The three spermathecae are elliptical and globular and are either transparent or white. They link the spermathecal glands and the middle oviducts through the spermathecal ducts. The spermathecal glands are thick and tubular and are also transparent. The spermathecal ducts can be observed passing through the spermathecal glands.

### 3.2. Developmental Processes of the Endo-Reproductive Organs of Adult H. illucens

#### 3.2.1. Development of the Testicle

The anatomical observation of the testicles of male flies from 1 to 10 days old was carried out, and the length and width of their testicles were measured and analyzed. The results show that the developmental morphology of the flies’ testicles varied with age (Figure 2). The testicles of the 1 d old male flies were translucent, with minimal yellow content. The yellow contents of the testicles increased gradually with increasing age, reaching a maximum at 5 days old (Figure 2A–C) and then gradually decreasing with time, while the yellow content gradually increased in the seminal vesicle, maybe due to the transfer of mature spermatozoa. In later stages, the testicles appeared translucent, and the yellow content accumulated more at the base of them and less at the end (Figure 2D,E).

Additionally, there were significant variations in the length (F = 5.264, *p* < 0.05) and width (F = 14.442, *p* < 0.05) of the adult testicles among the age groups (Figure 3). The length of the testicle tended to increase with age in the early stages of emergence, reaching a maximum length of 8.000 ± 0.27 mm at 3 days old. Thereafter, the length of the testicle began to decrease. There was a significant difference between the length of the testicle at 7 days old and that at 3 days old (*p* < 0.05).

The width of the testicles increased during the early adult stages, peaking at 3 days old at 0.552 ± 0.34 mm, and appeared to shrink considerably after 5 days old (Figure 4). The results show that the width of the testicle at 3 days old was significantly different from that at 6 days old (*p* < 0.05).

#### 3.2.2. Developmental Grading of the Ovaries

After emerging, the ovaries of female *H. illucens* gradually develops and mature. In a study by Zhang et al. [24], ovary development was classified into five stages based on ovary color, yolk deposition, and oogenesis levels. Based on the classification system of Zhang et al. [24] and the ovarian developmental dynamics in *H. illucens*, we systematically categorized the ovary development into five stages: Level I (previtellogenic developmental phase), Level II (yolk deposition phase), Level III (egg maturation phase), Level IV (peak phase of oviposition), and Level V (terminal phase of oviposition). Females in the previtellogenic developmental phase had smaller ovaries with no yolk deposited. The ovaries were flocculent and translucent, and the eggs were transparent, small, and rounded with tails (Figure 5A). The initiation of yolk deposition marked the transition from the ‘no yolk’ phase to the yolk deposition phase. During yolk deposition, the ovaries had began to turn yellowish, and the eggs were oval in shape with a hyaline tail at one end, increasing in size (Figure 5B). Eggs in this phase were not fully developed and the females had not begun to oviposit. The egg maturation phase is a transition phase between yolk deposition and the initiation of oviposition (Figure 5C). In this phase, eggs in the ovaries were arranged neatly with little gaps in between, and the ovaries turned yellow. During the egg maturation stage, eggs were fusiform, rapidly increasing in size, with a reduced hyaline caudal portion at one end, approaching maturity. Some oviposition occurred during the egg maturation stage. In the peak phase of oviposition, females began to deposit fully matured eggs regularly (Figure 5D). The eggs in the ovaries were rice-like, milky white, largest in size, and filled the ovaries. Oviposition completed in the terminal phase (Figure 5E). The ovaries were empty of mature eggs, constricted, and shriveled up. They contained a number of flocculent eggs that appeared to be of level I.

#### 3.2.3. Development of the Seminal Vesicle

*H. illucens* has three ellipsoidal spermathecae. Microscopic observation of the reproductive systems of females of all ages, both mated and unmated, revealed differences in the morphology and spermatheca contents. Specifically, the spermathecae in the mated females were filled with white, turbid content (Figure 6A). The spermatozoa were observed at as early as 2 days old in the spermathecae. A large number of elongated filamentous spermatozoon were observed in the contents under the microscope (Figure 6B), with spermatozoon twisting and curling around one another. By contrast, the spermathecae in the unmated females were transparent, with a small amount of content (Figure 6C), and no filamentous spermatozoon were detectable under the microscope (Figure 6D).

Meanwhile, the length and width of the spermathecae of mated and unmated adult females of all ages were measured and analyzed: after emergence, the length and width of the spermatheca increased with age and were significantly different between ages. The length and width of the spermathecae of mated and unmated females increased in the early adult phase (Figure 7 and Figure 8), and both reached their maximum at 5 days old. At 5 days of age, in mated females, the length of the spermatheca was 0.590 ± 0.024 mm, while the width of it was 0.500 ± 0.024 mm. In 5 d mated females, the length of the spermatheca was 0.588 ± 0.016 mm while the width of it was 0.427 ± 0. 016 mm. After 5 days old, the length and width of the spermatheca decreased in both mated and unmated females. At the same age, the length (F = 22.332, *p* < 0.05) and width (F = 385.570, *p* < 0.05) of the spermathecae of mated and unmated females were significantly different and the length and width of the spermathecae of mated females were significantly larger than those of unmated females at all ages.

### 3.3. Effects of Mating Status on the Egg Development Process

Anatomical observations of mated and unmated females aged 1–10 days old revealed that *H. illucens* egg morphology changes from round with a tail to oval with a tail, and then to spindly with a tail, and finally to a grainy rice shape. The shape changes corresponded to the different phases of ovary development. Mature eggs are rice-shaped. The development of eggs varied individually at the same age, and the proportion of eggs in different developmental phases varied in the ovaries of females of different ages (refer to Figure 9).

A comparison of egg development in mated and unmated females, based on the morphology at each day of age (Figure 10), revealed that the process of egg development was not synchronized in mated and unmated females. The eggs in the mated female flies begin to develop at 2 days old. On the third day, the yolk increased significantly in size and took an oval shape with a tail. By the fourth day, it was close to maturity and had a spindly shape with a tail. On the fourth day, mature eggs began to appear, accounting for 20% of the total. By the 6th day, 40% of mature eggs and 50% of eggs had already spawned. From the 7th to the 10th day of age, the percentage of mature eggs was less than 20%, indicating that the peak of egg maturity occurred on the 6th day of age. Most mated females had finished laying eggs by the 6th to the 10th day of age. The eggs from the unmated females exhibited a slower developmental rate during 1–3 days old, while also having lower levels of yolk deposition. Eggs tended to mature at the age of 4 days, presenting a spindly shape with a tail. There were almost no mature eggs in the ovaries of the unmated females until 5 days old, accounting for just 10%. The proportion of mature eggs at 6–10 days was still relatively low, accounting for 10–20%, and the eggs had a slightly rounded, shriveled shape.

## 4. Discussion

The endo-reproductive system of male *H. illucens* comprises the testicle, vas deferens, seminal vesicle, male accessory gland, and ejaculatory duct, while the female reproductive system is composed of the ovaries, spermatheca, spermathecal gland, oviduct, and female accessory gland. Compared to the endo-reproductive system of other insects, the accessory glands of *H. illucens* comprise one pair of tubular glands, while the accessory glands of *P. utilis* consist of 5–7 saccular glands. The structure of insect spermatheca can be classified into two types: that with specialized tubular spermathecal glands connected, as found in most advanced insects in Coleoptera, Hymenoptera, and Lepidoptera, and that with only glandular cells on the wall of the spermatheca [25], lacking separate spermathecal glands, as is found in lower insects such as the insects in Hemiptera [26], Dictyoptera (*Periplaneta americana* [27], *Ciulfina klassi* [28] and *Coptotermes gestroi* [29]) and Orthoptera [30]. In this study, we found that female *H. illucens* have three spermathecae and corresponding spermathecal glands, categorized as the former type. By contrast, *Bactrocera tau* [24] and *Bactrocera cucuribitae* [31] have two spermathecae and lack separate spermathecal glands, categorized as the latter type.

In general, the testicles are in the optimal condition at the time of sexual maturity of males, and they subsequently undergo gradual volume reduction. The reduction in the volume of the seminal vesicles may be related to the transfer of semen and the atrophy of the testicles. The negative correlation between testicular volume and age has also been confirmed in *Ostrinia nubilalis* [32] and the *Polygonia c-aureum* [33]. In locusts, it has been found that mating causes a large amount of semen to be discharged, thus causing a reduction in the size of the testicles. In this study, we found that the development of the testicle after emergence was obvious in adult *H. illucens* reared under laboratory conditions (27–29 °C, humidity: 50–60%). The analysis of the morphological development, length, and width of the testicle shows that the contents of the testicle were abundant around 2–5 days old and subsequently gradually decreased after 5 days old, and, at the same time, the content of the seminal vesicles gradually increased. Based on previous research [34] and the findings of this study, the optimal mating age of *H. illucens* is determined to be between 2 and 4 days old, with the mating rate decreasing after 6 days old. Therefore, we consider that the reduction in testicle size in the unmated male *H. illucens* after 5 d may be due to semen transfer and organ shrinkage. Based on the results, it can be hypothesized that the peak of spermatozoa development and maturation is around 2–5 days old, which may prompt male individuals to mate with females.

Additionally, the spermathecae of *H. illucens* had three saccular structures, with spermathecal glands wrapping around them. After mating, sperms are temporarily stored in the spermatheca [35]. Therefore, the presence or absence of sperm in the spermatheca was used in this study as a basis for determining whether females were mated or not. In addition, we found that the sperm was first found in the spermatheca at 2 days old, which is in agreement with a previous study which reported that *H. illucens* started to mate after 2 d of emerging [36].

The spermatheca is an organ that temporarily stores and maintains the activity of spermatozoon after insects copulate [35], and due to sperm storage, spermathecae enlarge [37]. The contents of the spermatheca consist of secretions and sperm, and some studies found that the size and secretion of the spermatheca increased with the time of mating [38,39]. The sperm contains spermatozoon and secretions from the male accessory glands [28]. The secretions from the male accessory glands in sperm can maintain the activity of spermatozoon and influence the egg-laying and sexual behavior of female flies [35]. The morphology of the spermatheca of mated and unmated females was significantly different. The spermathecae of unmated individuals were translucent with a lower content, whereas those of the mated females were white, turbid, and had more content, and the color depth changed with age, similarly to as described in *Tenuisvalvae notata* [40]. The mating status of female flies can be determined based on this phenomenon. Furthermore, a comparative analysis of the length and width of the spermatheca showed that the size of the spermatheca increased during the early adult stage of *H. illucens*, and then plateaued on about the 5th day after emergence. Conversely, the length and width of the spermatheca in mated females were significantly greater than those in unmated females across all ages. This discrepancy may be attributable to the filling and stimulation of sperm after mating, which leads to an enlargement of the spermatheca. In mated female *H. illucens*, the length and width of the spermatheca reached the maximum at 5 d. Most of the *H. illucens* mated at 2–5 d, and based on the percentage of mature eggs in the ovary at 1–10 d, we speculated that 6–7 d was the peak of spawning. In summary, we suppose that after 2–5 d of mating, semen is temporarily stored in spermatheca, the decrease in the volume of the spermatheca after 5 d is related to the release of sperm, the released sperm transfer for fertilization, and the females complete spawning at 6–7 d. In this study, it was also observed that the size of the spermatheca of unmated female *H. illucens* increased with age, and its length and width reached their maximum at 5 d, as in mated females. As da Silva et al. found in *Cryptotermes brevis* [39], the secretory activity of the spermatheca takes place prior to insemination, and as adult *H. illucens* mainly mate at 2–5 d, we suppose that if the secretory activity of the spermatheca was not limited by mating, it would continuously secrete before 5 d to prepare for the storage of semen. Such a condition has been reported in the unmated females of other insects, in which other factors influence the secretory activity of the spermatheca, rather than sperm transfer. Davey and Webster [41] found that in *Rhodnius prolixus*, the production of secretions is controlled by their neurosecretory cells. The decrease in the size of the spermatheca in unmated female *H. illucens* after 5 d may be due to other reasons, and the specific reasons for its atrophy are still unclear.

The morphological development of the ovary displayed distinct stages. Based on ovary color, yolk deposition level, and the level of egg development, Zhang et al. [24] categorized the ovary development of *Bactrocera tau* into five levels. Following Zhang et al. [24], we also categorized the ovarian development of *H. illucens* into five levels: Level I (previtellogenic developmental phase), Level II (yolk deposition phase), Level III (egg maturation phase), Level IV (peak phase of oviposition), and Level V (terminal phase of oviposition). Ovary development level can serve as a benchmark to determine sexual maturity [42]. Our study therefore serves as a reference for improving the mating rate, regulating the spawning behavior, and increasing the spawning output of the black soldier fly in captive breeding.

In this study, we found that mating status played a crucial part in the ovary development of *H. illucens*. The ovary maturity level of mated females (90% matured ovum at 6 days old) developed significantly faster than unmated individuals (10% at 10 days old). The secretions of the accessory gland in male *Locusta migratoria* [43] were identified to contain an oviposition-stimulating factor that can boost the maturation of eggs and the oviposition behavior of female flies. Similar findings have been reported in studies on *Helicoverpa armigera* [44], *Tribolium castaneum* [45] and Mosquitoes [46]. It can be speculated that the facilitating effect of mating on egg development may be related to the presence of male accessory secretions in the spermatheca. Furthermore, several hormones in insects, such as juvenile hormone (JH), can affect ovary development [47]. A previous study showed that JH can regulate ovary development via the control of glucose levels [48]. Further studies are needed to explore whether the faster development of the ovum in mated females is related to hormones, and whether interactions occur between hormones and the spermatozoa.

In summary, this study is a preliminary study of the composition of the endo-reproductive system and the developmental characteristics of the reproductive organs of *H. illucens*. It also investigates the connection between the ovary, the spermatheca, and the fertilization of eggs in adult females. Our observations show that unmated females barely spawn, which may be related to the mechanism by which female adults absorb nutrients from unfertilized eggs for life-sustaining purposes [34]. In this study, we only supplied the adults with water, and some studies have shown that providing adult H. illucens with artificial nutrients can extend their life and enhance their mating ability [49]. We propose a follow-up study of whether unmated female *H. illucens* can complete parthenogenesis when artificial nutrients are provided to the adults and will try to compare the oviposition, oviposition batch, and egg hatching of mated and unmated females to further explore the effects of mating on female *H. illucens*.

## Figures and Tables

**Figure 1 insects-16-00330-f001:**
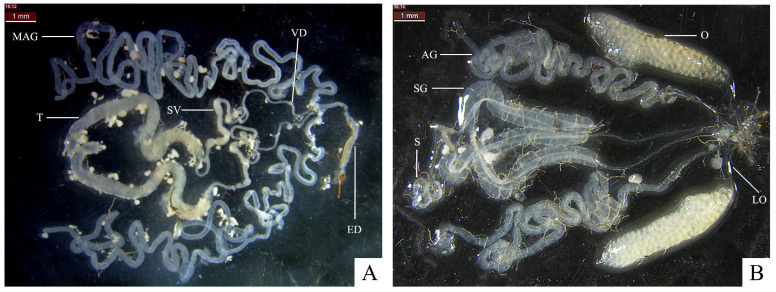
Endo-reproductive system of adult *H. illucens*. (**A**) Male endo-reproductive system; MAG: male accessory gland; T: testicle; SV: seminal vesicle; VD: vas deferens; ED: ejaculatory duct. (**B**) Female endo-reproductive system; AG: female accessory gland; LO: lateral oviduct; O: ovaries; S: spermatheca; SG: spermathecal gland.

**Figure 2 insects-16-00330-f002:**
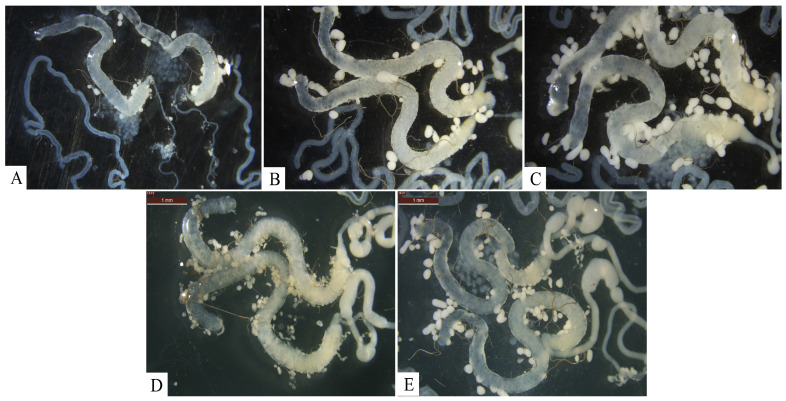
Developmental morphology of *H. illucens* testicles at different ages. (**A**–**E**) The testicles of 1, 3, 5, 7, 9 day old male *H. illucens*, respectively.

**Figure 3 insects-16-00330-f003:**
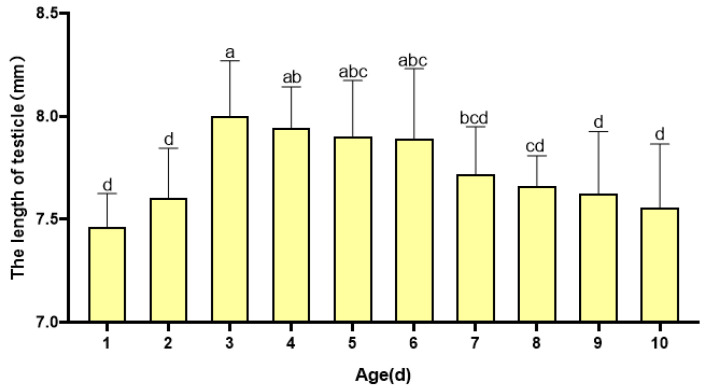
Changes in the length of the testicles of *H. illucens* at different ages. Data in the figure are average ± SD; different lowercase letters after data in the same column indicate significant differences at the 0.05 level (One-way ANOVA, Duncan method, *p* < 0.05).

**Figure 4 insects-16-00330-f004:**
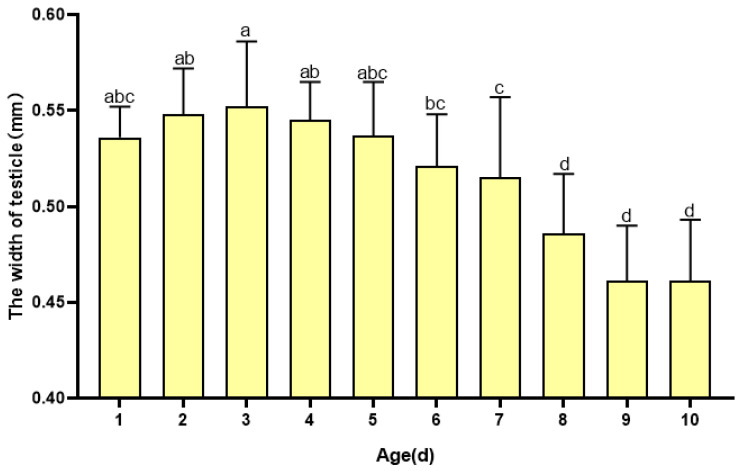
Changes in the width of the testicles of *H. illucens* at different ages. Data in the figure are average ± SD; different lowercase letters after data in the same column indicate significant differences at the 0.05 level (One-way ANOVA, Duncan method, *p* < 0.05).

**Figure 5 insects-16-00330-f005:**
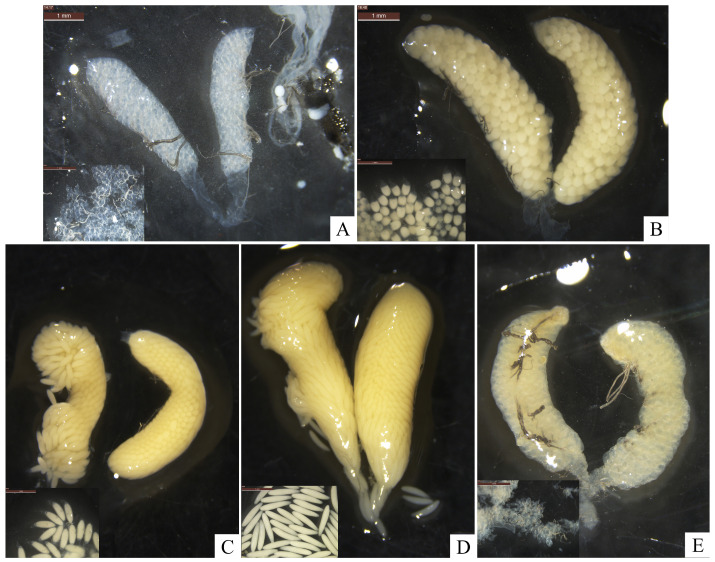
The morphology and classification of the ovaries of *H. illucens* at different developmental levels and corresponding egg developmental levels. (**A**) Level I, previtellogenic developmental phase; (**B**) Level II, yolk deposition phase; (**C**) Level III, egg maturation phase; (**D**) Level IV, peak phase of oviposition; (**E**) Level V, terminal phase of oviposition.

**Figure 6 insects-16-00330-f006:**
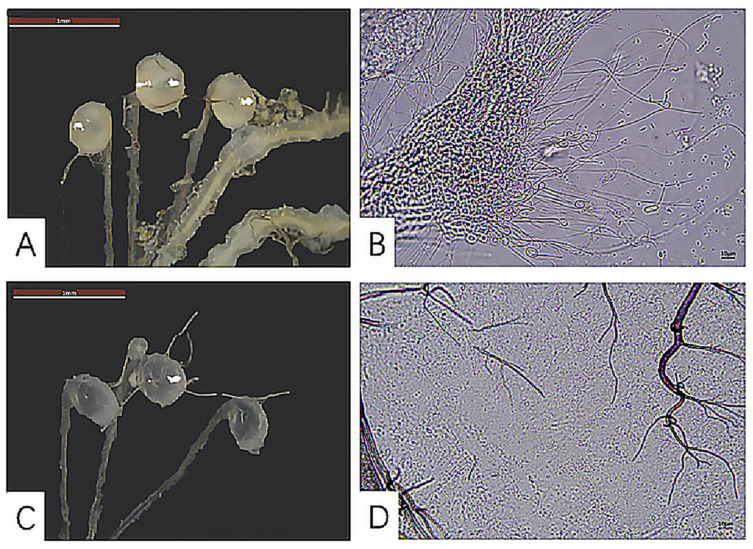
Comparison of the spermatheca morphology and contents between mated and unmated female adults of *H. illucens*. (**A**,**B**) The spermatheca and its contents in a mated female adult. (**C**,**D**) The spermatheca and its contents in an unmated female adult (2 days old).

**Figure 7 insects-16-00330-f007:**
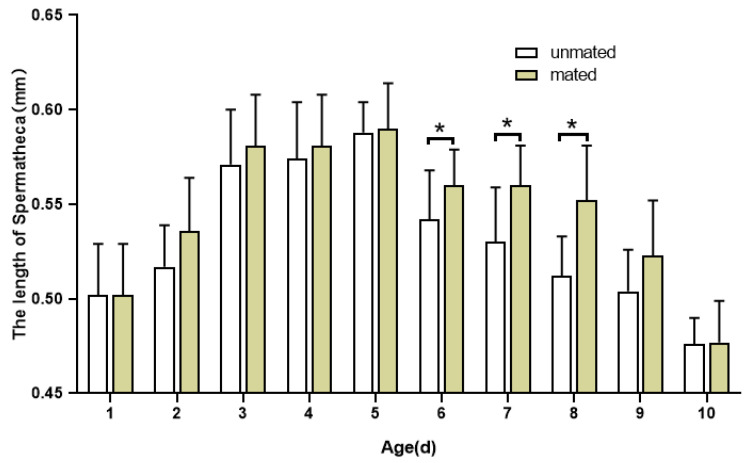
Changes in the length of the spermatheca between mated and unmated female adults of *H. illucens* at different ages. The asterisk * indicates that the testicle morphological parameters of different mating statuses at the same age are significantly different (Two-way ANOVA, Duncan’s method, *p* < 0.05).

**Figure 8 insects-16-00330-f008:**
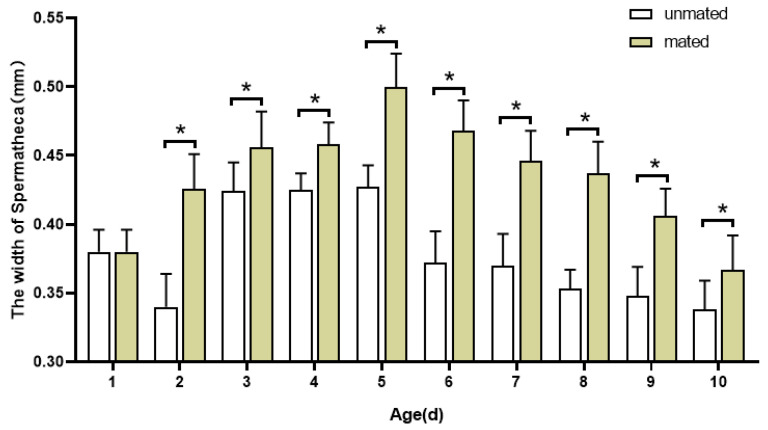
Changes in the width of the spermatheca between mated and unmated female adults of *H. illucens* at different ages. The asterisk * indicates that the testicle morphological parameters of different mating statuses at the same age are significantly different (Two-way ANOVA, Duncan’s method, *p* < 0.05).

**Figure 9 insects-16-00330-f009:**
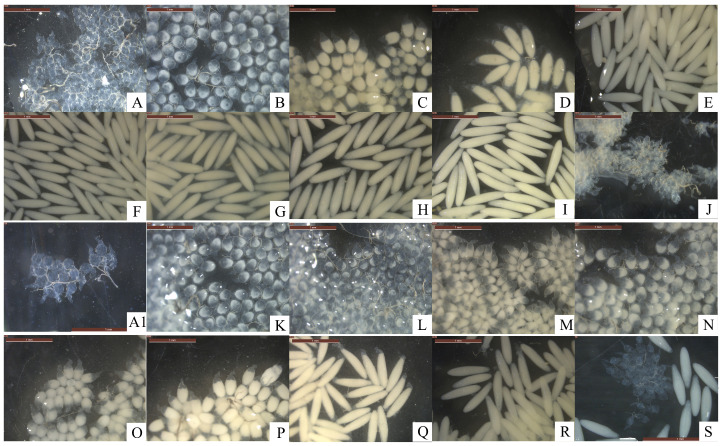
Comparison of egg development at different ages between mated and unmated females of *H. illucens*. (**A**) The 1-day-old female eggs (in a mixed cage). (**A1**) The 1-day-old female eggs (in a separate cage). (**B**–**J**) Mated female eggs at 2–10 days old. (**K**–**S**) Unmated female eggs at 2–10 day old females.

**Figure 10 insects-16-00330-f010:**
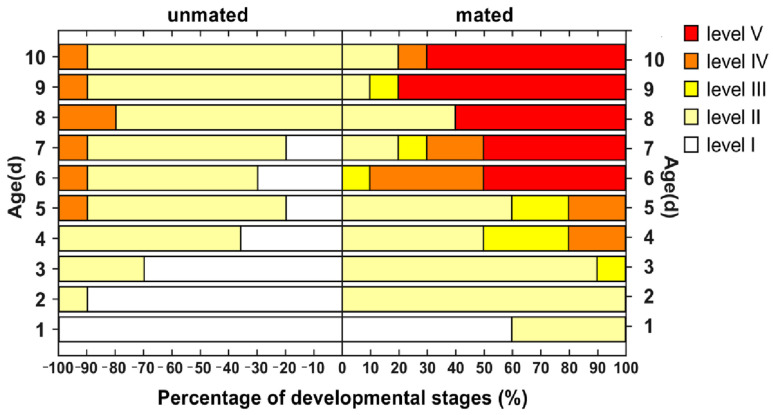
Ovarian development of mated and unmated female *H. illucens* at different ages.

## Data Availability

The data presented in this study are available on request from the corresponding author.

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
