# Peer review of "Development of the Endo-Reproductive System and the Effect of Mating Status on Egg Development in Adult Hermetia illucens L."

_insects, 2025, doi:10.3390/insects16040330_

Round 1

Reviewer 1 Report

Comments and Suggestions for Authors

Dear authors,
I consider that the work is interesting and provides important data for the knowledge of the species. I recommend to review the wording, especially in the materials and methods section, because it is not clear how many individuals of each sex were used in total and whether the males were only unmated or mated.
Do you have any information about the yellow content of the testicles that you mention (line 152)? Are there spermatozoa?
In the results section they must solve the problem that they talk about sperm libraries when referring to males (lines 163 -164 and 170-171).

Author Response

  1. L97-99: In the materials and methods section, because it is not clear how many individuals of each sex were used in total and whether the males were only unmated or mated.

Response: Thanks for the valuable suggestion. The detailed information has been added to L02-107.

  1. L106-108: were the individuals alive? if not, describe the method used to kill the individuals.

Response: Thanks for the valuable suggestion. The treatment method of test insects has been added to L111-113.

  1. L157-160: I suggest that the references be grouped first, those of image A and then those of image B so that it is easier to interpret.

Response: Thanks for the valuable suggestion. The references have been grouped in L162-165.

  1. L152-153: Was this evaluated in unmated males? What is the yellow content cited in the text? What could be the cause of this decrease?

Response: Thanks for your valuable suggestion. The evaluation was in unmated males. The yellow content here consist mainly of mature spermatozoa. In this study, the reduction of the yellow contents in testicles may be due to the transfer of mature spermatozoa, and an increase in the yellow contents of the Seminal vesicle can be observed. We added the discussion in the L174-176.

  1. L180&L188: Does the male's reproductive system have a spermathecae? where is is?

Response: We apologize for the careless error. We have corrected the wrong word in L185 and 193.

  1. L197-199: Add that the categorization is based on the work of Zhang et al. (2018) because the categories were not established by you?
    Response: Thanks for yourvaluable suggestion. The related citation hasbeen added in L201-204.

7.what are the tails?

Response: Thanks for your valuable suggestion. The sharper photo has been added to the Fig.5 A on L223.

  1. L238-241: Are you referring to the lengh of the spermatheca of unmated females?

Response: Thanks for your valuable suggestion. We have changed this part to a clearer expression in L244-247.

  1. L271&273:what are egg particles? i do not unterestad the difference between eggs and eggs particles?

Response: Thanks for your valuable suggestion. We apologize for the careless mistake. We have removed inappropriate expressions

  1. L287: Does this mean that the oocytes from the same ovary do not all mature at the same time? or that some oocytes do not mature? It would be interesting if this were clarified since either of the two situations is relevant information.

Response: Thanks for your valuable suggestion. We apologize for the careless mistake. We have corrected the note. Figure 10 shows the ovarian development at different ages of mated and unmated female H. illucens. The situation you describe does exist, insect egg development is a complex process (Xu et al., 2002), which is influenced by hormone, nutritional status, temperature and humidity (Wheeler, 1996). In Helicoverpa armigera, eggs at the base of the ovary are more mature than those at the end (Zhang, 2013). In Ostrinia furnacalis and Spodoptera exigua, some of the eggs start to develop at the end of the pupae stage, in the early stage of emergence, there are already some semi-mature eggs in the adults. Whether there is the case in H. illucens still requires further study and may be carried out in subsequent experiments.

Reviewer 2 Report

Comments and Suggestions for Authors

Dear Editor and authors. I’ve finished my review on “Development of the endo-reproductive system and effect of mating status on the egg development in adult Hermetia illucens L.” by Chen et al. Overall, the work is well-presented and easy to follow. I have some comments, especially on the methods and interpretation of the data, which require a careful attention of the authors before acceptance.

Introduction

A link between the last two paragraphs is missing. What was the purpose of the study? Is it just to provide morphological and morphometrical descriptions of the reproductive system in black soldier flies or did the authors also aim to relate these features with the reproductive biology of the species? Please give more details.

The methods are well described. I just have a few comments as follows.

L86 – Is there a reason why you did not include mated males? I was wondering if mating would also affect testicular development.

I’m unsure if length and width are the best parameters for testicles (and maybe the spermatheca) due to their cylindrical/rounded shapes. I would suggest measuring the superficial area using the ImageJ freehand tool or similar software. Since the images are of high quality, the authors could easily extract these measurements and provide a precise drawing of the testicular/spermathecal development.

Results

Figures are well shown. The dissections were very precise and the authors should be congratulated for that.

L143 and figures: Avoid the term FAG because this word might be offensive in English. I suggest AG for the female accessory glands.

L153-155: “Testicles” are mentioned three times in the same sentence. You should revise the text to avoid this kind of issue. Moreover, there are several cases of double space or double points/commas.

L160: How do you explain the reduction in the size of the testicles? Is it related solely to sperm transfer or as individuals age, they suffer a shrinkage of these structures? For how long do male and female adults of H. illucens live? If there is a correlation between the “window for reproduction” and the dimensions of reproductive structures, it must be discussed.

L205: The title states seminal vesicles but the content refers to the spermathecae. Please fix the information.

L206-227: Spermathecae likely increased in size due to sperm storage. Interestingly, the size and secretory activity increase with mating (Pascini et al., 2012, 2013; da Silva and Costa-Leonardo, 2024). Sperm counting and histochemical tests would be valuable to support the data presented here. The authors noted variation in the size between different-age unmated females. How do you explain that? Is there increasing size in response to other events rather than mating? Maybe the unmated females were somehow exposed to stimuli from males, even though there was no direct contact.

Pascini, T. V., Ramalho-Ortigão, M., & Martins, G. F. (2012). Morphological and morphometrical assessment of spermathecae of Aedes aegypti females. Memórias do Instituto Oswaldo Cruz107, 705-712.

Pascini, T. V., Ramalho-Ortigäo, J. M., & Martins, G. F. (2013). The fine structure of the spermatheca in Anopheles aquasalis (Diptera: Culicidae). Annals of the Entomological Society of America106(6), 857-867.

da Silva, I. B., & Costa-Leonardo, A. M. (2024). Mating- and oviposition-dependent changes of the spermatheca and colleterial glands in the pest termite Cryptotermes brevis (Blattaria, Isoptera, Kalotermitidae). Protoplasma261(2), 213-225.

L240: How would you relate egg development and spermatheca size? For instance, a reduction in spermathecal dimensions might be related to decreasing the number of stored sperm since they are used to fertilize eggs. Did you record the number of eggs laid aiming to propose such a correlation?

Figure 9 does not read well. In the way it is, it looks like the figures go from E to K rather than to F. Please replace the order.

Discussion: How are your results on the endo-reproductive system of black soldier flies related to their biology? The discussion includes many comparisons to other insect taxa but it does not explore how the morphometrical data might somehow relate to the reproductive success of males and females. I suggest the authors improve this section following these guides.

L280: The lack of spermathecal glands and the occurrence of a secretory epithelium is a common feature of large clade Dictyoptera (cockroaches, termites, and mantises) as well and must be referred to as such (Lawson and Thomas, 1970; Winnick et al., 2009; da Silva and Costa-Leonardo, 2023).

Lawson, F. A., & Thomas, J. C. (1970). Ultrastructural comparison of the spermathecae in Periplaneta americana (Blattaria: Blattidae). Journal of the Kansas Entomological Society, 418-434.

Winnick, C. G., Holwell, G. I., & Herberstein, M. E. (2009). Internal reproductive anatomy of the praying mantid Ciulfina klassi (Mantodea: Liturgusidae). Arthropod structure & development38(1), 60-69.

da Silva, I. B., & Costa‐Leonardo, A. M. (2023). Mating mediates morphophysiological changes in the spermathecae of Coptotermes gestroi queens. Entomologia Experimentalis et Applicata171(5), 361-373.

L282: You first mention B. tau and B. cucuribitae here so the scientific names must be provided without abbreviations. Are these species of fly as well? It is not clear why you’ve compared your data to these.

285-291: You repeat the results here rather than comparing them to other data. I suggest suppressing most of the information.

L305: The correct is spermatozoa (plural)

349: Do you expect to find parthenogenesis in this species? Your present data on the developmental stages of the eggs strongly suggest that those from unmated females do not reach stage V. So how would they be viable?

Author Response

  1. Introduction, A link between the last two paragraphs is missing. What was the purpose of the study? Is it just to provide morphological and morphometrical descriptions of the reproductive system in black soldier flies or did the authors also aim to relate these features with the reproductive biology of the species? Please give more details.

Response: Thanks for your valuable suggestion. We have added a detailed description on L78-80.

  1. L86:Is there a reason why you did not include mated males? I was wondering if mating would also affect testicular development.

Response: Thanks for your valuable suggestion. In this study, we focused on the effects of mating on the endo-reproductive system of female H. illucens, whether mating has an effect on the development of the endo-reproductive system of male H. illucens has not been studied and and may be followed up in subsequent studies.

  1. I’m unsure if length and width are the best parameters for testicles (and maybe the spermatheca) due to their cylindrical/rounded shapes. I would suggest measuring the superficial area using the ImageJ freehand tool or similar software. Since the images are of high quality, the authors could easily extract these measurements and provide a precise drawing of the testicular/spermathecal development.

Response: We sincerely appreciate your insightful suggestions regarding morphological assessment. The selection of linear measurements (length/width) as baseline parameters was predicated on their cylindrical/rounded morphological configurations of the structures, which facilitates intuitive dimensional characterization. While acknowledging the scientific validity of surface area quantification for evaluating developmental dynamics, we have incorporated this recommended methodology into our experimental protocol for subsequent investigations.

  1. L143 and figures: Avoid the term FAG because this word might be offensive in English. I suggest AG for the female accessory glands.

Response: Thanks for your valuable suggestion. We apologize for the careless error. We have corrected the offensive word.

  1. L153-155: “Testicles” are mentioned three times in the same sentence. You should revise the text to avoid this kind of issue. Moreover, there are several cases of double space or double points/commas.

Response: Thanks for your valuable suggestion. We have corrected the inappropriate expression.

  1. L160: How do you explain the reduction in the size of the testicles? Is it related solely to sperm transfer or as individuals age, they suffer a shrinkage of these structures? For how long do male and female adults ofH. illucenslive? If there is a correlation between the “window for reproduction” and the dimensions of reproductive structures, it must be discussed.

Response: Thanks for your valuable suggestion. We've added the related discussion on L174-176, L310-316, L321.

  1. L205: The title states seminal vesicles but the content refers to the spermathecae. Please fix the information.

Response: Thanks for your valuable suggestion. We apologize for the careless error. We have corrected the wrong word.

  1. L206-227: Spermathecae likely increased in size due to sperm storage. Interestingly, the size and secretory activity increase with mating (Pascini et al., 2012, 2013; da Silva and Costa-Leonardo, 2024). Sperm counting and histochemical tests would be valuable to support the data presented here. The authors noted variation in the size between different-age unmated females. How do you explain that? Is there increasing size in response to other events rather than mating? Maybe the unmated females were somehow exposed to stimuli from males, even though there was no direct contact.

Response: Thanks for your valuable suggestion. We have added this section on L359-371.

  1. L240: How would you relate egg development and spermatheca size? For instance, a reduction in spermathecal dimensions might be related to decreasing the number of stored sperm since they are used to fertilize eggs. Did you record the number of eggs laid aiming to propose such a correlation?

Response: Thanks for your valuable suggestion. Related discussions have been added to L349-355, and we are sorry that we did not record the number of eggs laid in this study, it may be studied in subsequent experiments.

  1. Figure 9 does not read well. In the way it is, it looks like the figures go from E to K rather than to F. Please replace the order.

Response: Thanks for your valuable suggestion. We have replaced it with a more appropriate image.

  1. Discussion: How are your results on the endo-reproductive system of black soldier flies related to their biology? The discussion includes many comparisons to other insect taxa but it does not explore how the morphometrical data might somehow relate to the reproductive success of males and females. I suggest the authors improve this section following these guides.

Response: Thanks for your valuable suggestion. We have improved the disscussion as suggested on L349-356.

  1. L280: The lack of spermathecal glands and the occurrence of a secretory epithelium is a common feature of large clade Dictyoptera (cockroaches, termites, and mantises) as well and must be referred to as such

Response: Thanks for your valuable suggestion. We have added this section on L304-305.

  1. L282: You first mentionB. tauand B. cucuribitae here so the scientific names must be provided without abbreviations. Are these species of fly as well? It is not clear why you’ve compared your data to these.

Response: Thanks for your valuable suggestion. We have corrected the scientific names. We have not found other relevant studies on other insects in Stratiomyidae, the Bactrocera tau, Bactrocera cucuribitae and H. illucens both belong to the Diptera, the endo-reproductive systems of insects in the same order may have some similarity structural characteristic, so we compared them here.

  1. L285-291: You repeat the results here rather than comparing them to other data. I suggest suppressing most of the information.

Response: Thanks for your valuable suggestion. We have deleted part of the information here.

  1. L305: The correct is spermatozoa (plural)

Response: Thanks for your valuable suggestion. We apologize for the careless error. We have corrected the wrong word.

  1. L349: Do you expect to find parthenogenesis in this species? Your present data on the developmental stages of the eggs strongly suggest that those from unmated females do not reach stage V. So how would they be viable?

Response: Thanks for your valuable suggestion. Egg development is related to nutrient availability, the adult H. illucens do not feed. In this study, we only supplied the adults with water, some studies have shown that providing adult H. illucens with artificial nutrients can extend their life and enhance mating ability[51]. We propose a follow-up study, whether unmated female H. illucens can complete parthenogenesis when artificial nutrients are provided for the adults and try to compare the oviposition, oviposition batch, and egg hatching of mated and unmated females to further explore the effects of mating on the female H. illucens.

[51] Wang, C., Ye, X. M., Du, J. et al. Effect of different CP/ME ratios diets on growth performance of Hermetia illucensFeed research, 2023, 46(08):84-89.

Round 2

Reviewer 2 Report

Comments and Suggestions for Authors

Dear editor and authors,
After my second review, I believe the manuscript is ready for publication as the authors performed substantial changes in the text and figures. The images are of high quality and this study might be very welcome by the scientific community. 
I have just two appointments: the references 28 and 40 are the same. Please merge them into one; I left comments about issues with the english language that must be fixed before the final acceptance by the Editor.

Comments on the Quality of English Language

I would suggest the authors to revise the english of the phrases and sections that were included after the first round of comments, as some of them are incongruent. I do not believe it is going to need a third round of review as the authors can easily fix these issues.

Author Response

1.The references 28 and 40 are the same. Please merge them into one; I left comments about issues with the english language that must be fixed before the final acceptance by the Editor.

Response: Thanks for the valuable suggestion. We have removed the duplicated references and corrected the incongruent words.